# Implementation of Pharmacogenetics in First-Line Care: Evaluation of Its Use by General Practitioners

**DOI:** 10.3390/genes14101841

**Published:** 2023-09-22

**Authors:** Denise van der Drift, Mirjam Simoons, Birgit C. P. Koch, Gemma Brufau, Patrick Bindels, Maja Matic, Ron H. N. van Schaik

**Affiliations:** 1Department of Clinical Chemistry, Erasmus MC University Medical Center, 3015 GD Rotterdam, The Netherlands; 2Department of Hospital Pharmacy, Erasmus MC University Medical Center, 3015 CN Rotterdam, The Netherlands; 3Department of Clinical Chemistry, Result Laboratory, 3318 AT Dordrecht, The Netherlands; 4Department of General Practice, Erasmus MC University Medical Center, 3015 GD Rotterdam, The Netherlands

**Keywords:** pharmacogenetics, pharmacogenomics, GP, clinical implementation, CYP450, personalized medicine

## Abstract

Pharmacogenetics (PGx) can explain/predict drug therapy outcomes. There is, however, unclarity about the use and usefulness of PGx in primary care. In this study, we investigated PGx tests ordered by general practitioners (GPs) in 2021 at Dept. Clinical Chemistry, Erasmus MC, and analyzed the gene tests ordered, drugs/drug groups, reasons for testing and single-gene versus panel testing. Additionally, a survey was sent to 90 GPs asking about their experiences and barriers to implementing PGx. In total, 1206 patients and 6300 PGx tests were requested by GPs. *CYP2C19* was requested most frequently (17%), and clopidogrel was the most commonly indicated drug (23%). Regarding drug groups, antidepressants (51%) were the main driver for requesting PGx, followed by antihypertensives (26%). Side effects (79%) and non-response (27%) were the main indicators. Panel testing was preferred over single-gene testing. The survey revealed knowledge on when and how to use PGx as one of the main barriers. In conclusion, PGx is currently used by GPs in clinical practice in the Netherlands. Side effects are the main reason for testing, which mostly involves antidepressants. Lack of knowledge is indicated as a major barrier, indicating the need for more education on PGx for GPs.

## 1. Introduction

Pharmacogenetics is the field of analyzing genetic polymorphisms in order to explain or predict drug therapy outcomes. Mostly, this is focused on analyzing single-nucleotide polymorphisms (SNPs) in genes encoding drug-metabolizing enzymes and/or drug transporters. The rationale behind using DNA information to guide drug therapy lies in the fact that only 30–60% of patients respond properly to treatment with antidepressants, antipsychotics, β-blockers and statins [1], and that 5–7% of hospitalizations are due to adverse drug reactions (ADRs) [2,3,4]. Having DNA information available can help to reduce adverse drug reactions by 30%, as was recently demonstrated in the PREPARE study [5]. Guidelines on how to dose drugs based on genetic information are currently available for several gene–drug combinations (www.cpicpgx.org, accessed on 1 August 2023). In total, there are now 28 CPIC guidelines available, addressed in 43 publications: antidepressants/*CYP2D6* and *CYP2C19* [6,7]; SSRIs/*CYP2D6* and *CYP2C19* [8]; SSRIs/*CYP2D6*, *CYP2C19*, *CYP2B6*, *SLC6A4* and *HTR2A* [9]; codeine/*CYP2D6* [10,11]; opioids/*CYP2D6*, *OPRM1* and *COMT* [12]; atomoxetine/*CYP2D6* [13]; statins/*SLCO1B1*, *ABCG2* and *CYP2C9* [14,15,16]; warfarin/*CYP2C9* and *VKORC1* [17,18]; clopidogrel/*CYP2C19* [19,20,21]; proton pump inhibitors (PPI)/*CYP2C19* [22]; odansetron/*CYP2D6* [23]; tamoxifen/*CYP2D6* [24]; voriconazol/*CYP2C19* [25]; non-steroidal anti-inflammatory drugs (NSAIDs)/*CYP2C9* [26]; fluoropyrimidines/*DPYD* [27,28]; efavirenz/*CYP2B6* [29]; phenytoin/*CYP2C9* and *HLA-B* [30,31]; rasburicase/*G6PD* [32,33]; tacrolimus/*CYP3A5* [34]; carbamazepine/*HLA-B* [35,36]; abacavir/*HLA-B* [37,38]; allopurinol/*HLA-B* [39,40]; PEG-interferon α/*IFNL3*(*IL28B*) [41]; aminoglycosides/*MT-RNR1* [42]; volatile anesthetic agents and succinylcholine/*RYR1* and *CACNA1S* [43]); thiopurines/*TPMT* (and *NUDT15*) [44,45,46]); atazanavir/*UGT1A1* [47]; and ivacaftor/*CFTR* [48]. In addition, the FDA has published a table of pharmacogenetic associations (https://www.fda.gov/medical-devices/precision-medicine/table-pharmacogenetic-associations, accessed on 1 August 2023), divided into three sections: (1) associations for which the data support therapeutic management recommendations; (2) associations for which the data indicate a potential impact on safety or response; and (3) associations for which the data demonstrate a potential impact on pharmacokinetic properties only. The CPIC guidelines and the FDA association table do not, however, overlap completely [49]. The Dutch Pharmacogenetics Working Group (DPWG), a working group of the Royal Dutch Society for Advancing Pharmacotherapy (KNMP: Koninklijke Maatschappij ter bevordering der Pharmacie), saw the first publication of their guidelines in 2008 [50], with an update in 2011 [51]. Currently, the DPWG has 108 recommendations (www.pharmgkb.org, accessed on 1 August 2023), many of which were also published separately with the background information on how the advice was reached: SSRIs/*CYP2D6* and *CYP2C19* [52], antipsychotics/*CYP2D6*, *CYP3A4* and *CYP1A2* [53], irinotecan/*UGT1A1* [54], fluoropyrimidines/*DPYD* [55], atomoxetine and methylphenidate/*CYP2D6* and *COMT* [56], opioids/*CYP2D6* [57], allopurinol/*ABCG2* and *HLA-B* [58]), and folic acid and methotrexate/*MTHFR* [58]. In the Netherlands, these guidelines are in use and every pharmacist has access to genotype-based dosing recommendations. Based on the increased interest and availability of guidance documents, an increasing number of laboratories are nowadays offering pharmacogenetic testing.

As to the position of pharmacogenetic testing in the healthcare setting, there has been debate about its position in first-line care. Whereas pharmacogenetics undergoes adequate implementation now in second-line care for a small group of high-risk medications (clopidogrel/*CYP2C19*, abacavir/*HLA-B*5701*, and capecitabine/*DPYD*), the majority of medicines are being prescribed in primary care. In 2015, 27 out of the 80 medicines for which the DPWG had guidelines available at the time were regularly prescribed in first-line care [59]. Several publications on the potential of pharmacogenetic testing in first-line care have been published, also addressing pharmacist-initiated testing for GPs [60,61]. A recent study, estimating the potential impact of implementing pre-emptive pharmacogenetic testing in primary care in the UK, demonstrated that 19–21% of all new prescriptions for 56 drugs (5.2–5.7 million prescriptions/year in total) were, in fact, involving an actionable gene–drug interaction, according to CPIC and DPWG guidelines, showing the high potential of PGx testing in primary care [62]. A comparable study, performed in the Netherlands, reported 23.6% (n = 856,002) of all new prescriptions for 45 drugs given by GPs as actionable [63]. An initiative to identify potential barriers to implementing pharmacogenetics among primary care physicians in the Netherlands, however, was not successful, due to low response rates [64]. At Erasmus MC at the Dept. Clinical Chemistry, we have been offering clinical diagnostic testing for pharmacogenetics since 2005, being an expert center recognized by the International Federation of Clinical Chemistry (IFCC) since 2007. Starting in 2005, we have seen a growth in the number of test requests from 25 tests in 2005 up to 23,284 tests in 2021, representing 8500 patients in total.

The aim of this present study was to analyze the number of requests sent in by general practitioners (GPs), and we subsequently used our real-world clinical data to identify the genes, as well as the medications, of interest. Furthermore, we wanted to investigate the reasons for requesting a PGx test. Finally, we were interested in experiences and barriers so we distributed a questionnaire to GPs in our academic GP network associated with Erasmus MC (named “PrimEUR”), to ask them about their experiences and barriers for using PGx in primary care.

## 2. Materials and Methods

In the period January 1 to December 31, 2021, 8500 PGx request forms in total, representing 8500 patients, were received at Dept. Clinical Chemistry, Erasmus MC, representing 23,284 test requests. Erasmus MC offers 26 individual genes for testing, but also allows standardized panel testing, including DNA panel—basic; DNA panel—extended; a psychiatry panel; cardiac panel; pain panel and oncology panel (Table 1). The mean turnaround time was 7 working days, except for individual *CYP2C19*/clopidogrel testing requests, which had a turnaround time of 24 h. Clinicians could choose whether to send in blood samples, or have a buccal swab kit sent directly to their patients.

The GP survey was conducted among 90 GPs from the Erasmus MC Academic GP network PrimEUR, and sent out by email in February 2022, with a reminder after one week. The GPs were not selected based on whether or not they had experiences with PGx testing. The survey contained 13 questions and was divided into questions regarding characteristics of the participating GP, the application of PGx and the sources that were used to obtain information about PGx. The survey contained multiple choice questions, but also included an “other…” option. The survey closed 3 weeks after the first invite. Analyses were performed using IBM SPSS Statistics 25 for Windows. Descriptive statistics were used to summarize each question. 

## 3. Results

### 3.1. Materials, Request Forms and Genes Requested

From the total of 8500 patients sent to Erasmus MC for PGx testing, 1281 patients (15%) were sent in by GPs, either directly (76%) or through specific GP laboratories (24%). For analysis, 75 test request forms were excluded: 8 forms were not available, 12 were duplicates and 55 forms were not sent in on behalf of a different physician, leaving 1206 patients (14%), representing 6301 out of 23,284 test requests (27%). From the 1206 request forms obtained from primary care, 65% were sent in as buccal swabs and 35% as blood samples. The total number of individual tests was 6301, resulting in an average of 5.2 tests/patient. The top two genes were *CYP2C19* (17%) and *CYP2D6* (16%) (Figure 1a). PGx panels were requested in 795/1206 patients (66%), representing 5257/6301 tests (83%). Among these, 43% were for the DNA panel—extended; 24% were for the psychiatry panel; and 15% were for the DNA panel—basic (15%) (Figure 1b). Individual gene testing (so outside of a panel) was highest for *CYP2D6* (272/978; 27.9%) and *CYP2C19* (206/895; 23.7%), and lowest for *VKORC1* (2/493; 0.4%), *OPRM1* (3/91; 3.3%), *ABCB1* (2/64; 3.1%) and *SLCO1B1* (16/510; 3.1%). 

### 3.2. Drugs

The majority of GPs (706/1206; 59%) listed on the request form which drugs (groups) the PGx test was requested for: 146 (12%) listed both the drug and drug group, 505 (42%) listed only the drug and 55 (4.5%) listed only the drug group. In total, 237 different drugs were mentioned on the requested forms, representing 30 different drug groups. PGx testing was most often requested for clopidogrel, at 23% (147/639), followed by citalopram (8%), amitriptyline (6.8%) and metoprolol (6.8%). In total, 50 drugs were mentioned in at least 1% of requests (Table 2).

Regarding drug groups, antidepressants were most frequently mentioned (70% of forms; 850/1206), followed by antihypertensive (26%) and antiplatelet drugs (26%) (Figure 2).

### 3.3. Reasons for Requesting PGx Testing

Reasons for requesting PGx testing, in 70% of patients (850/1206), were as follows: (a) low drug plasma concentrations, (b) high drug plasma concentrations, (c) side effects, (d) non-response, (e) prior to therapy, or (f) other. On 195/1206 forms (16%), more than one indication was given. The most frequent reason for requesting a PGx test was “side effects”, as stated in 79% of request forms (669/846), followed by “non-response” (27%) and “prior to therapy” (13%). In addition to the standard choices, in 2.8% of request forms (24/857), “PGx-variant in family member” was mentioned (Figure 3).

### 3.4. Results of the Survey on PGx

The survey was returned by 29 out of 90 GPs, resulting in a response rate of 32%. Most responders (20/29; 69%) were working in a group practice/healthcare center. Regarding their work experience, 2/29 (7%) worked for 0–5 years as GPs, 14/29 (48%) worked for 5–20 years and 13/29 (45%) worked for more than 20 years. Self-reported familiarity with PGx ranged from “slightly familiar” (11/29; 38%) and “moderately familiar” (7/29; 24%) to “familiar” (10/29; 34%) and “very familiar” (1/29; 3%). None of the GPs reported being “unfamiliar” with PGx testing. As to how the GPs learned about PGx, most respondents reported a refresher course/magazine (20/29; 69%) and patient cases (13/29; 45%). In total, 16/29 respondents (55%) had requested a PGx test in the last year, with the majority (10/16; 63%) requesting a basic DNA passport, followed by requests for *CYP2C19* (7/14; 50%) and *CYP2D6* (6/14; 38%). On the question regarding the barriers experienced in using PGx, 59% of GPs reported “unclear when to test”, “cost for patient” (48%) and “lack of experience” (45%). Other barriers reflected unclarity in testing or test interpretation (Figure 4). 

Regarding the sources consulted by GPs with regard to PGx, pharmacists were mentioned mostly as the main source (34%), followed by publications of the Dutch College of General Practitioners (NHG) (21%) and the website www.farmacogenetica.nl (accessed on 1 August 2023) (17%). The laboratory offering PGx testing, the pharmacology compendium ‘Farmacotherapeutisch Kompas’ and the GP genetics website (www.huisartsengenetica.nl, accessed on 1 August 2023) each scored 10% for consulted sources. GPs did not use information from the drug label, PharmGKB, CPIC, DPWG scientific literature or the KNMP PGx database.

## 4. Discussion

Our results indicate that PGx testing is being taken up by GPs and has a place in clinical practice, both through the total number of patients (1281) and the number of test requests (6301) in 2021. It is important to note that the GPs ordering these test were GPs from all over the Netherlands, and were not restricted to Erasmus MC or GPs from the Erasmus MC-associated PrimEUR Network. Buccal swab testing (75%) was clearly preferred above blood sampling (25%), compared to the percentage of 20% of buccal swabs when taking all PGx test received at our department in 2021 into account. There is, in addition, a preference by GPs for requesting multiple tests per patient, resulting in an average of 5.2 tests/patient. This is higher than the number of hospital-based test requests, which have an average of 2.3 tests/patient (Dept. Clinical Chemistry, PGx test request data 2021). Most tests ordered by GPs were requested inside a panel, ranging from 72% for *CYP2D6* to 99.6% for *VKORC1*. Apparently, statin use itself was not regarded as a specific reason for requesting a PGx test, as seen in the relatively low number of requests for individual *SLCO1B1* tests. It cannot be excluded, however, that side effects of statins triggered PGx panel testing. Interestingly, DNA panel—extended was the most highly requested panel test (43%). This, in our opinion, could reflect that, when considering a PGx test, a total overview is preferred over a smaller, perhaps more focused panel by patients and/or GPs. Alternatively, this could also reflect the limited experience of GPs in terms of which test to order.

Although from individual drugs, clopidogrel was most often mentioned (23%), on a group level, antidepressants represented the major reason (51%) for requesting a PGx test. The other drug groups in the top seven were antihypertensives, antiplatelet drugs, opioids, statins, β-blockers and proton pump inhibitors. In total, 30 different drug groups were indicated. From the individual drugs, it is interesting that of the drugs mentioned in >1% of request forms, almost half (27/50) do not have a direct indication for PGx testing based on DPWG guidelines. This may possibly reflect that communication with physicians on when PGx testing can be helpful, should be improved. Otherwise, it could reflect that physicians want to provide all relevant information on the medication a patient is on. When a non-guideline drug was the only reason for asking for PGx testing, no test was performed. In other instances, the gene was tested for which an appropriate test and corresponding dosing guideline was available. The medication mentioned on the form was, however, still included in our overview.

As for reasons for PGx testing, side effects was by far the most reported (79%). This reflects the Dutch GP guideline that indicates that PGx testing should be considered only after side effects are experienced, or ineffective therapy, and not prior to therapy. Yet, 13% of forms still indicated “prior to therapy” as the reason. Since we did find combinations where both “side effects” and “prior to therapy”, or “non-response” and “prior to therapy” boxes were checked, it seems that this question was multi-interpretable. Indeed, after side effects (or non-response), a new drug is to be prescribed, meaning that both options on the form could be indicated. This also caused the total percentage for this question to exceed 100%. High/low plasma drug concentrations were hardly mentioned as a reason for requesting PGx testing (2.8%).

Regarding the survey among GPs, the response rate of 30% (29/90) was above expectations, since an earlier study on Dutch GPs was based on only eight interviews, which was reported as a very low response rate [64]. The GPs that were approached by us were part of an academic network (PrimEUR), which is why the response rate may have been higher in our case. The distribution of working experience in the interviewed GPs of the PrimEUR group ranged from relatively new (7%) to more than 30 years of experience (14%), and was quite balanced over the different years of experience. Most GPs were to some extent, familiar with PGx, with only one reporting “very familiar”. Most respondents had asked for a PGx test in the last year, reflecting the answers on familiarity. Consulting a pharmacist, refresher courses, magazines, Dutch literature, patient cases and congresses were reported as main sources of knowledge on PGx, whereas international literature scored 0%. As to barriers experienced, lack of knowledge on when to test (59%), which test to request (41%), how to order a test (34%), or lack of knowledge on how to interpret test results (31%) stood out, clearly reflecting the need for more information for GPs on how to use PGx. 

As a second argument, concerns with costs for the patient (48%), or unclarity in terms of reimbursement (10%), were mentioned as barriers. In the Netherlands, PGx testing is reimbursed by insurance companies when requested by a GP, provided there is a valid reason for requesting the test, this usually being the occurrence of side effects or non-response. However, each patient has a maximum amount that he/she needs to contribute each year out of pocket, which is usually around EUR 385, meaning that there may still be costs for the patient when this maximum amount is not yet reached. Because of this policy, Erasmus MC experiences a doubling of the number of PGx panel test requests in the last 3 months of the year, when most patients have reached this maximum through their own contribution, and PGx testing does not result in extra costs for the patient. It is important to notice that the GPs in the PrimEUR group, because of their affiliation with academia, are not representative of the average GP. The fact that in this selected group a lack of knowledge is mentioned as a barrier to implementing PGx, again stresses the importance of providing more tools and education for GPs on PGx.

In a recent study performed in the US, comparing the attitudes of healthcare providers to PGx testing, it was shown that primary care providers overwhelmingly perceived PGx to be useful [65], as is the case in our own experience with teaching GPs. In fact, primary care providers were more likely to rate PGx testing as useful compared to specialty care providers [65], indicating that PGx is a tool that fits the needs of GPs more than those of specialty care providers. In a recent survey among thirty-four clinicians in the US that had minimal experience with PGx and PGx resources, it was shown that the expectation among participants was that PGx indeed has the ability to improve medication-related outcomes, but that many lack the confidence to apply PGx results in their practice [66]. This again indicates that GP education on PGx is of high importance. In this regard, pharmacist-initiated PGx testing could be of value, as revealed by van der Wouden et al. [61] via an investigation on the feasibility and real-world impact of this approach. In their prospective study, they provided community pharmacists the opportunity to pre-emptively request a panel of eight pharmacogenes to guide drug dispensing with a clinical decision support system for 200 primary care patients. Pre-emptive testing would relieve the pressure on GPs contemplating which PGx test should be ordered, since results are already available. In a mean 2.5-year follow up in this study, it was shown that PGx panel reports were successfully recorded in 96% of pharmacists, and in 67% of GP electronical medical records. In 97% of cases, this enabled patients to (re)use PGx panel results at least once. In total, 24.2% of prescriptions had actionable drug–gene interactions. The authors concluded that pre-emptive panel-based PGx testing is feasible and that real-world impact is believed to be substantial in primary care. These findings confirmed the findings of an earlier study in the Netherlands on the estimated nationwide impact of implementing a pre-emptive PGx panel approach to guide drug prescribing in primary care, where it was shown that 23.6% of all new prescriptions were, in fact, drug–gene interactions and that dose adjustments or switching to another drug would follow in 5.4% of all new prescriptions [63].

## 5. Conclusions

PGx testing does have a role in GP practice, and is indeed also already applied in the Netherlands by this group of healthcare practitioners. There is a preference for buccal swabs as a testing material over blood sampling. The occurrence of side effects is most prominently mentioned as the reason for requesting PGx testing, and the drug group mostly involved is antidepressants. Lack of knowledge on PGx is still indicated as a major barrier to implementing PGx in primary care, stressing the need for developing additional tools and education for GPs on PGx.

## Figures and Tables

**Figure 1 genes-14-01841-f001:**
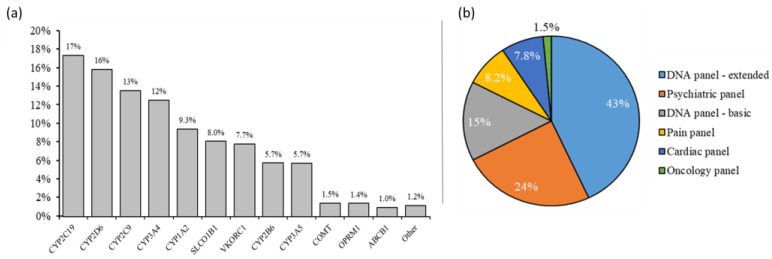
PGx tests requested by GPs. (**a**) Distribution per gene, including both separate and panel testing requests (n = 6301); displayed are the genes above a 1% threshold. PGx tests requested, but not reaching a 1% threshold, were *BChE*, *CYP2C8*, *CYP3A7*, *DPYD*, *GADL*, *HLA-B*1502*, *HLA-A*3101*, *HLA-B*5701*, *MTHFR*, *NUDT15*, *TPMT*, *UGT1A1* and *UGT1A9*; (**b**) distribution on requested panel testing (n = 795).

**Figure 2 genes-14-01841-f002:**
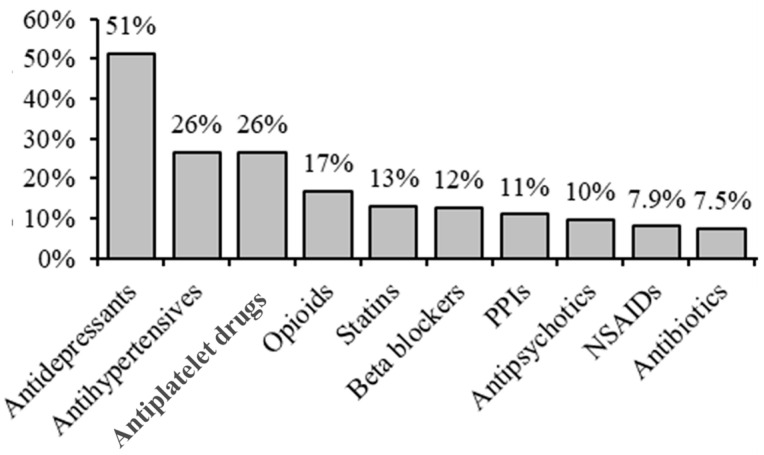
Top 10 drug groups for which PGx testing was requested, based on 706 forms. PPIs: proton pump inhibitors; NSAIDs: non-steroidal anti-inflammatory drugs.

**Figure 3 genes-14-01841-f003:**
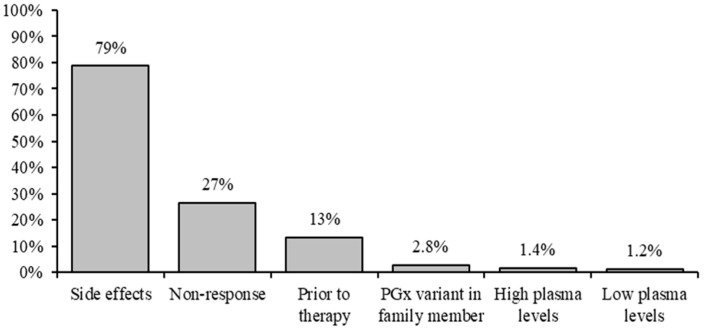
Main reasons for requesting PGx testing, based on 850 forms.

**Figure 4 genes-14-01841-f004:**
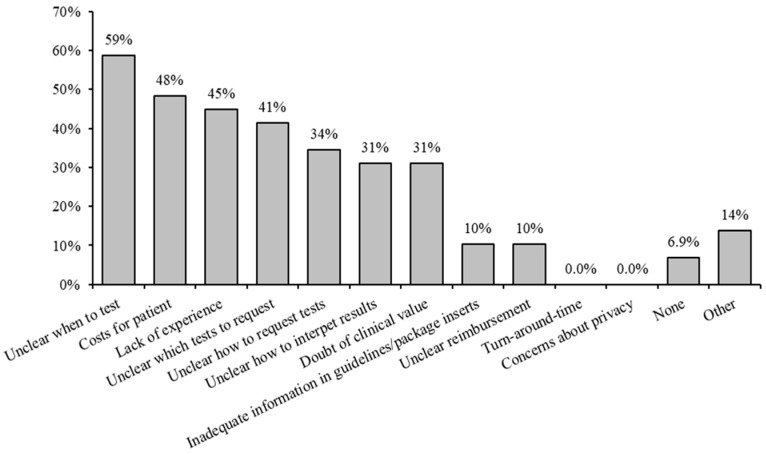
Barriers to applying PGx experienced by respondents (n = 29).

**Table 1 genes-14-01841-t001:** Available PGx tests and panel tests at Erasmus MC in 2021.

PGx Panel Name	PGx Tests
DNA panel—basic	*CYP2C9*, *CYP2C19*, *CYP2D6*, *CYP3A4*, *VKORC1*, *SLCO1B1*
DNA panel—extended	Is DNA panel—basic *+ CYP1A2*, *CYP2B6*, *CYP3A5*
Psychiatry panel	*CYP1A2*, *CYP2C9*, *CYP2C19*, *CYP2D6*, *CYP3A4*
Cardiac panel	*CYP2C9*, *CYP2C19*, *CYP2D6*, *VKORC1*, *SLCO1B1*, *ABCB1*
Pain panel	*CYP2C9*, *CYP2D6*, *CYP3A4*, *OPRM1*, *COMT*
Oncology panel	*CYP2D6*, *CYP3A4*, *CYP3A5*, *DPYD*
Individual tests	*CYP1A2*, *CYP2B6*, *CYP2C8*, *CYP2C9*, *CYP2C19*, *CYP2D6*, *CYP3A4*, *CYP3A5*, *CYP3A7*, *BChE*, *DPYD*, *TPMT*, *NUDT15*, *UGT1A1*, *UGT1A9*, *ABCB1*, *ABCC2*, *ABCG2*, *SLCO1B1*, *HLA-A*3101*, *HLA-B*1502*, *HLA-B*5701*, *HLA-B*5801*, *HLA-B*1511*, *GADL1*, *VKORC1*

**Table 2 genes-14-01841-t002:** Drugs mentioned by GPs on request forms (n = 706); threshold >1%. Drugs for which the DPWG have no guidelines or no adapted dosing recommendations available are indicated in italics.

Drug	%	Drug	%	Drug	%
Clopidogrel	23.0%	*Lisinopril*	3.2%	Clomipramine	1.8%
Citalopram	8.0%	Rosuvastatin	2.9%	*Metformin*	1.7%
Amitriptyline	6.8%	Atorvastatin	2.9%	*Bisoprolol*	1.7%
Metoprolol	6.8%	*Fluoxetine*	2.9%	*Verapamil*	1.5%
Tramadol	6.3%	Quetiapine	2.8%	*Amoxicillin*	1.5%
Sertraline	6.1%	*Morphine*	2.8%	*Bupropion*	1.5%
Pantoprazole	5.7%	*Pregabalin*	2.6%	*Ac. Salic. Acid*	1.5%
Venlafaxine	5.4%	Risperidone	2.5%	*Duloxetine*	1.4%
*Amlodipine*	5.4%	*Diclofenac*	2.5%	*Valsartan*	1.4%
Simvastatin	4.9%	*Ezetimibe*	2.2%	*Diazepam*	1.2%
Oxycodone	4.6%	*Paracetamol*	2.2%	*Perindopril*	1.2%
Paroxetine	4.5%	Aripiprazole	2.0%	*Naproxen*	1.1%
Omeprazole	4.3%	*Nifedipine*	2.0%	*Ibuprofen*	1.1%
Nortriptyline	4.1%	*Methylphenidate*	2.0%	*Enalapril*	1.1%
Escitalopram	3.7%	*Lidocaine*	2.0%	*Fentanyl*	1.1%
*H.chlorothiazide*	3.5%	Codeine	1.8%	*Oxazepam*	1.1%
Mirtazapine	3.5%	*Losartan*	1.8%		

## Data Availability

Data are contained within the article.

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
