# Peer review of "Implementation of Pharmacogenetics in First-Line Care: Evaluation of Its Use by General Practitioners"

_genes, 2023, doi:10.3390/genes14101841_

Round 1
Reviewer 1 Report
In the present review, Denise Van der Drift et al address the actual matter of PGx in the clinical practise. It is time to apply in translational medicine the huge amount of data obtained by the several international and multicentre/single centre investigations.
The review is perfectly written and the discussed topics are appropriate.
Few minor points should be further considered.
Abstract.
-First line (line 15): Please delete -(partly)- : Pharmacogenetics (PGx) can explain/predict drug therapy outcomes.
-Line 20: In total, 1,206 patients and 6,300 PGx test were requested by GPs, representing 14% of total patients and 27% of total tests. Please explain better the two different reported percentages (14% and 27%) as done afterward in the text of the manuscript; did you mean “respectively”?
-Line 26: ..Side effects are the main reason for testing, and mostly involves antidepressants... Please add one additional result (i.e. the second one in term of percentage for both “main reason for testing” and the category of drug).
Introduction.
Line 37: …only 30-60% of patients respond properly to treatment.. Please be more precise by adding details.
Do you have sex-disaggregated data? It should be extremely interesting to stratify data by sex, or at least add information in the introduction and in the discussion sections.
Results.
Please disclose PPIs and NSAIDs in the text and figure/tables.
Discussion.
Line 197 and line 216; authors stated that:
Most tests ordered by GPs were requested inside a panel, ranging from 72% for CYP2D6 to 99.6% VKORC1.
and afterward they add,
PGx testing should be considered only after side effects, or ineffective therapy, and not prior to therapy. Yet, 13% of forms still indicated as reason “prior to therapy”. Since we did find combinations where both “side effects”, and “prior to therapy”, or “non-response” and “prior to therapy” boxes were checked, it seems that this question was multi-interpretable.
Accordingly, please cite the pharmacogenetics studies suggested below in which a comprehensive approach has been applied to also decide the drug initial dosage and the dosage maintenance (in that case the anticoagulant warfarin) assessed by testing CYP2C9 and VKORC1 haplotypes using the WRI score (i.e. Warfarin responsive index):
-PLoS One. 2016 Sep 8;11(9):e0162084. doi: 10.1371/journal.pone.0162084.
-J Hum Genet. 2006;51(3):249-253. doi: 10.1007/s10038-005-0354-5.
Finally, one additional interesting drug/molecule has not been considered in the present review, as well as another important class of disease: e.g. Mayzent (Siponimod, Novartis) in the treatment of selected neurodegenerative disease (i.e. multiple sclerosis progression). Siponimod is a sphingosine-1-phosphate (S1P) receptor modulator, and the drug label requires the common CYP2C9 genotype, contraindicating its use in patients with the CYP2C9*3/*3 genotype. Even if Authors do not have their own data, this novel finding can be briefly discussed and considered.
Please cite: Biomed Pharmacother. 2022 Sep;153:113536. doi: 10.1016/j.biopha.2022.113536
Author Response
All questions of the reviewer were addressed: please see the attachement.

Reviewer 2 Report
Article
Implementation of pharmacogenetics in first line care: evaluation of its use by general practitioners
Denise van der Drift , Mirjam Simoons , Birgit CP Koch , Gemma Brufau , Patrick Bindels , Maja Matic and Ron HN van Schaik*
Summary:
The study investigated PGx test ordered by general practitioners (GP) in 2021 at the Dept. Clinical Chemistry, Erasmus MC, and analyzed gene test ordered, drugs/drug groups, reason for testing and single gene versus panel testing. Additionally, a survey was sent to 90 GPs asking their experiences and barriers for implementing PGx.
General concept comments:
The manuscript is scientifically sound and the experimental design is clear and presented in a well-structured manner. The cited references are recent publications and relevant. The manuscript does not include an excessive number of self-citations. The tables are relevant, they correctly show the data in most cases; data is interpreted clearly in most cases. Some tables need improvement. Conclusions are consistent with the evidence and arguments presented. The conclusion gives clinical relevance to improve the translational value of the research.
Specific comments and suggestions for Authors:
78. Please write DPWG instead of “DWPG”!
Please increase the clarity of Table 1.: DNA panel basic and extended seem to be a little confusing, however data are appropriate!
98. What is “PrimEUR”?
121. „From the total of 8,500 patients sent to Erasmus MC for PGx testing, 1,281 patients 121 (15%) were sent in by GPs.” How many GPs indicated this number of tests? Are they all GPs from PrimEUR?
149. Table 2. „Drugs for which DPWG has no guidelines or no adapted dosing recommendations available, are indicated in italics” – Do unnecessary tests generate extra costs for the GPs or the patients?
152. „anticoagulant 26%” – Which drugs do you mean under anticoagulant? Clopidogrel is not anticoagulant but antiplatelet agent. Anticoagulants are warfarin, acenocoumarol, however they are not listed in Table 2.
The same question is in Figure 2.: In case of clopidogrel the drug group antiplatelet agent is correct.
Was there any indication for VKORC1, because of anticoagulant use?
149. vs. 155.
How many request forms were analyzed (651 vs 706)?
157.
Is in GP practice relevant/possible to measure drug plasma level in the Netherlands? If yes, by which drugs? Is the reason for low PGx test based on the fact that drug plasma level measurement is unavailable?
166.
How did you select the 90 GP to whom you sent the survey?
188.
What does KNMP mean?
223.
High/low plasma drug concentrations were hardly mentioned as a reason for requesting PGx testing (2.8%). – Is it possible to measure drug plasma concentration in GP practices?
247
“Q4” – Please clarify!
Author Response

(The authors gave the same response as above.)
